# Differences in Treatment Outcomes and Prognosis between Elderly and Younger Patients Receiving Definitive Radiotherapy for Cervical Cancer

**DOI:** 10.3390/ijerph17124510

**Published:** 2020-06-23

**Authors:** PeiYu Hou, ChenHsi Hsieh, MingChow Wei, ShengMou Hsiao, PeiWei Shueng

**Affiliations:** 1Department of Radiation Oncology, Far Eastern Memorial Hospital, New Taipei City 220, Taiwan; jcgv03@gmail.com (P.H.); chenciab@gmail.com (C.H.); 2Faculty of Medicine, School of Medicine, National Yang-Ming University, Taipei 112, Taiwan; 3Institute of Traditional Medicine, School of Medicine, National Yang-Ming University, Taipei 112, Taiwan; 4Departments of Obstetrics and Gynecology, Far Eastern Memorial Hospital, New Taipei City 220, Taiwan; wei@mail.femh.org.tw (M.W.); smhsiao2@gmail.com (S.H.)

**Keywords:** cervical cancer, elderly, radiotherapy

## Abstract

The aim was to compare the clinical outcomes and prognostic factors of cervical cancer between elderly and younger women, and to explore which treatment strategy is more appropriate for elderly patients. We retrospectively reviewed patients with cervical cancer receiving definitive radiotherapy (RT) between 2007 and 2016, and divided them into two age groups: age < 70 vs. age ≥ 70. The clinical outcomes were compared between the two age groups. The median follow-up was 32.2 months. A total of 123 patients were eligible, 83 patients in group 1 (age < 70), and 40 patients in group 2 (age ≥ 70). Patients in group 2 received less intracavitary brachytherapy (ICRT) application, less total RT dose, and less concurrent chemoradiotherapy (CCRT), and tended to have more limited external beam radiotherapy (EBRT) volume. The treatment outcomes between the age groups revealed significant differences in 5-year overall survival (OS), but no differences in 5-year cancer-specific survival (CSS), 66.2% vs. 64.5%, and other loco-regional control. In multivariate analyses for all patients, the performance status, pathology with squamous cell carcinoma (SCC), International Federation of Gynecology and Obstetrics (FIGO) stage, and ICRT application were prognostic factors of CSS. The elderly patients with cervical cancer had comparable CSS and loco-regional control rates, despite receiving less comprehensive treatment. Conservative treatment strategies with RT alone could be appropriate for patients aged ≥ 70 y/o, especially for those with favorable stages or histopathology.

## 1. Introduction

Cervical cancer was the fourth most frequently occurring female’s cancer in 2018, with an estimated 570,000 new cases according to the recently published report of the World Health Organization (WHO), and was the fourth leading cause of cancer-related death in 2018, with an estimated 311,000 deaths. In developing countries, it was the second most commonly diagnosed cancer and the third leading cause of cancer-related death [1]. For cancer epidemiology in the currently-aged society, there is an increasing proportion of elderly patients who have been diagnosed. Recently, more than 15% of cervical cancer cases were found in women aged over 65 years in the United States, as reported by the American Cancer Society in 2018 [2].

Either surgery for early stage cervical cancer, or definitive concurrent chemoradiotherapy (CCRT) for patients with locally advanced cervical cancer was the mainstay treatment policy as a worldwide guideline recommendation, supported by randomized control trials published by the large serial Gynecologic Oncology Group (GOG) and Radiation Therapy Oncology Group (RTOG) [3,4,5]. Adjuvant radiotherapy (RT), with or without concurrent chemotherapy, for patients with pathological adverse risk factors provided better outcomes, as was shown by GOG 92 and GOG 109 trials [6,7,8]. However, in these large clinical studies, only a few elderly patients were included. In a review article that noticed the underrepresentation of elderly patients in clinical trials, out of more than 17,000 publications in PubMed until 2016, only 24 publications reported the clinical status of elderly patients with cervical carcinoma [9].

The prognosis of the elderly population with cervical cancer was generally thought to be inferior to that of the younger population, regardless of an increased morbidity rate, mortality rate, and rate of treatment complications. There was a retrospective study that specifically focused on the management of elderly patients, and provided data showing that treatment compliance and completion rates were much lower in the elderly population with surgery, RT or chemotherapy as the standard indicated policy [10]. In a Japanese study, some elderly patients experienced severe toxicity, although radiotherapy is generally effective for them [11]. These conditions might contribute to the adverse effect on the prognosis of elderly patients [12]. Limited historical data showed inconsistent results indicating that that elderly patients might have poor prognosis [13,14], or similar survival rates, when compared with younger patients [15,16].

The current study compared clinical outcomes and prognostic factors between elderly and younger women with cervical cancer undergoing definitive RT treatment in Asia. The results of this study will allow gynecological and radiation oncologists to determine which treatment strategy is more appropriate for elderly patients, and to provide more precise medical management for them.

## 2. Materials and Methods

The study retrospectively reviewed patients with cervical cancer receiving definitive RT at the Far Eastern Memorial Hospital (FEMH) in Taiwan between 2007 and 2016. The inclusion criteria were as follows: pathologically proven cervical carcinoma, locally advanced disease or patients who were medically intolerant to surgery and received definitive RT, concurrent chemotherapy with RT was prescribed as indicated or by the investigator’s choice. The exclusion criteria were patients without pathology proven malignancy, or those who received surgery as primary treatment, or those who received palliative care only. All the included patients were divided into two age groups: group 1 of age < 70, and group 2 of age ≥ 70. The clinical outcomes and prognostic factors were analyzed between the groups. The human experimentation committee of the Far Eastern Memorial Hospital approved the retrospective data analysis (FEMH-107003-E).

### 2.1. RT

The RT treatment protocol consisted of external beam radiotherapy (EBRT) with or without subsequent brachytherapy in definitive settings. The radiation volumes of EBRT included gross tumor volume (GTV) as gross tumor and/or gross lymphadenopathy (LAP) if present. The clinical target volume (CTV) covered the GTV with sufficient vaginal margin from the gross tumor, parametrium, and involved lymphatic regions of the whole pelvis including the common iliac, external iliac, internal iliac, obturator, and presacral regions. Extended field EBRT to cover the paraaortic region was allowed if there was clinically/pathologically pelvic LAP(s) and/or paraaortic LAP(s). The planning target volume (PTV) was defined as CTV plus a 5–10 mm margin for setup error. The EBRT delivered 45 to 50.4 Gy in 25 to 28 fractions with a daily fraction throughout the 5–6 weeks. An additional 10 to 15 Gy boost dose to a gross tumor and/or gross LAP(s) would be allowed. The treatment volume of EBRT could be adapted only to gross disease without an elective lymphatic region if patients were intolerant as a physician’s choice. The treatment planning of EBRT was adopted in 3D conformal fields, intensity-modulated radiation therapy (IMRT), volumetric-modulated arc therapy (VMAT), or tomotherapy techniques. At least 100% of the CTV volume was to receive 100% of the prescription dose, and at least 95% PTV volume was to receive 100% prescription dose. The radiation dose constraints of organs at risk (OARs) met the criteria requirement as usual. Image guidance, such as mega-voltage computed tomography (MVCT) or cone-beam computed tomography (CBCT), was optionally used to define the anatomy and internal soft tissue positioning. Concurrent chemotherapy during EBRT was given if necessary. After completing EBRT, subsequent intracavitary brachytherapy (ICRT) was delivered with the high dose-rate (HDR) remote after-loading technique, using the radioactive source Iridium-192 (Ir-192). The prescription dose of ICRT was 25–30 Gy in 5 fractions to point A according to the International Commission of Radiation Units (ICRU) 38 report, in which 1–2 fractions were performed per week. The brachytherapy dose was converted from HDR to low dose-rate (LDR) at a ratio of 4/3, and the total radiation dose was the sum of the EBRT dose and converted brachytherapy dose.

### 2.2. Chemotherapy

If the patient received concurrent chemotherapy with RT, the regimen consisted of weekly cisplatin 30–40 mg/m^2^ or, alternatively, carboplatin or paclitaxel was administered if there was poor renal function.

### 2.3. Follow-Up

Patients underwent follow-up evaluations with a clinical examination and pelvic magnetic resonance imaging (MRI), or a computed tomography (CT) imaging survey, every 3 months for the first 2 years, every 6 months for the subsequent 3 to 5 years, and then every 12 months thereafter.

### 2.4. Outcomes

The clinical outcomes estimated overall survival (OS), cause-specific survival (CSS), loco-regional recurrence-free survival (LRRFS), and local recurrence-free survival (LRFS). The survival time was measured from the diagnosis date.

### 2.5. Statistical Analyses

The patients’ characteristics were compared between the two age groups using independent Student’s *t*–tests for the continuous variables, and chi-squared tests for the categorical variables. The Kaplan–Meier method was used to estimate the survival curve, and to compare the survival curve with the Log-Rank test. The differences in the results were considered statistically significant when the *p* value was less than 0.05. A Cox proportional hazards model was used for multivariable analysis of the prognostic factors. The Statistical Product and Service Solutions (SPSS) Statistics Version 22.0 (IBM Corps., Armonk, NY, USA) was used for statistical analyses.

## 3. Results

### Patient Characteristics

There were 123 patients with cervical cancer eligible from January 2007 to December 2016. The median age was 62 years old (range: 25–92). The majority of histopathology was squamous cell carcinoma (SCC) in 108 patients. The International Federation of Gynecology and Obstetrics (FIGO) stage included stage I to stage IV, mostly with stage IIB to IIIB. Among them, 83 patients (67.5%) were divided into group 1 (age < 70), and the other 40 patients (32.5%) were divided into group 2 (age ≥ 70). The histopathology, FIGO stage, EBRT volume, and overall treatment time (OTT) of RT were not significantly different between the two groups. Elderly patients seemed to have more proportional advanced disease from FIGO IIIA to IVA, and favored receiving a limited EBRT volume confined to gross disease only, without elective lymphatic region (15% vs. 3.6%, *p* = 0.06) compared with the younger patients, although it was a nonsignificant trend.

The elderly patients indeed had a relatively poor performance status, they received less total RT dose combining EBRT and brachytherapy (70.3 Gy vs. 80.4 Gy, *p* = 0.002) compared with group 1 patients, less brachytherapy use (32.5% vs. 69.9%, *p* < 0.001), and less concurrent chemotherapy use (about half of them received CCRT, compared with 89.2% of group 1, *p* < 0.001). Overall, the old age group had a more conservative treatment strategy. The details of the patient characteristics were presented in Table 1.

The median follow-up was 32.2 months. The treatment outcomes between groups 1 and 2 revealed that the median OS time was not reached in group 1 vs. 37 months in group 2, and the 3-year and 5-year OS rates were 63.9% vs. 50.6% and 60.4% vs. 39.1%, respectively (*p* = 0.006). Patients younger than 70 years of age had better OS than elderly patients and achieved significant differences. However, the other clinical results of CSS, LRRFS, and LRFS were all comparable between the two groups as shown: 3-year and 5-year CSS were 68.2% vs. 70.9% and 66.2% vs. 64.5% (*p* = 0.89), 3-year and 5-year LRRFS were 82.8% vs. 82.1% and 78.2% vs. 82.1% (*p* = 0.75), 3-year and 5-year LRFS were 85.3% vs. 82.1% and 82.6% vs. 82.1% (*p* = 0.91), respectively. The curves of OS, CSS, LRRFS and LRFS are displayed in Figure 1.

These treatment outcomes were listed in Table 2.

No differences were found in the CSS, LRRFS and LRFS rates between the two groups, although reduced OS was noted in elderly patients, with an approximately 10–20% decrease from that of the younger patients.

We evaluated the prognostic factors using univariate and multivariate analysis methods. On univariate analysis for all eligible patients, the Eastern Cooperative Oncology Group Performance Status (ECOG) 0–2, pathology with SCC, FIGO stage, total RT dose, CCRT and ICRT application had significant effects on CSS, with *p* values <0.001, <0.001, <0.001, 0.045, 0.013 and 0.02, respectively. However, age, EBRT target volume, and OTT were not significant factors of CSS with *p* values of 0.89, 0.64, and 0.58, respectively. The detailed contents were shown in Table 3.

On multivariate analysis for all patients, the ECOG 0–2 (*p* = 0.003, hazard ratio (HR) 0.14, 95% confidence interval (CI): 0.038–0.523), pathology with SCC (*p* = 0.001, HR 0.26, 95% CI: 0.115–0.573), FIGO stage (*p* < 0.001, HR 2.32, 95% CI: 1.487–3.626), and ICRT application (*p* = 0.045, HR 0.49, 95% CI: 0.242–0.983) persisted as significant factors of CSS. Other factors including age, total RT dose, CCRT, EBRT volume, and OTT revealed no significant effect on CSS.

## 4. Discussions

The current study explored any differences in treatment results and prognostic factors between elderly and younger populations. It appeared that elderly patients had comparable clinical outcomes, except in OS. The 3-year and 5-year OS of patients younger than 70 was 63.9% and 60.4%, and is comparable with the treatment result of the Surveillance, Epidemiology, and End Results (SEER) database, which was 66.2% between 2008 and 2014 [17], and is similar to the 60% to 70% rate of estimated 5-year net survival in many high-income countries [18]. For group 2 patients aged 70 years or older, the 3-year and 5-year OS rates were 50.6% and 39.1%, respectively, and the 3-year and 5-year CSS rates were 70.9% and 64.5%, respectively. In historically published reports or review articles, 3–5 year OS ranged from 33% to 89%, and 3–5 year CSS ranged from 58% to 80% for aging patients with cervical cancer at least older than 55 y/o [19]. The treatment quality for patients in our study, either in the general population or the elderly group, achieved comparable results with previous data and worldwide recognition.

In our study, patients aged ≥ 70 had worse OS—an approximately 10–20% reduction of OS compared to that of patients aged < 70. However, patients aged ≥ 70 could achieve similar treatment results in CSS, LRRFS, and LRFS even though they were subject to a relatively conservative treatment policy, which included a lower proportion of CCRT, less brachytherapy use, a lower total RT dose, and a trend to favor limited EBRT volume. This result was compatible with previous studies showing equivalent disease-specific survival for patients aged ≥ 65 but worse OS compared with those aged < 65 [20].

The analysis of prognostic factors in our study revealed that ECOG 0–2, pathology with SCC, FIGO, and ICRT had significant effects on CSS, both in univariate analysis and multivariate analysis. The performance status had been reported as a significant prognostic factor for OS and PFS [21,22]. Our data also confirmed the prognostic value of performance status on CSS. As a result of the multivariate analysis of our study, age, concurrent chemotherapy use, extended EBRT volume, or higher total RT dose were not shown to independently influence outcomes. Only 52.5% of group 2 patients had CCRT in our review, while others received RT alone. The conservative treatment option was commonly found in other studies [14,23]. In a multicenter, retrospective, registry-based study with a large number of cervical cancer patients aged ≥ 61 with advanced stage IIB–IV, the aged group was less likely to receive CCRT (odds ratio 0.31; 95% CI 0.20–0.48) compared with patients aged < 60 y/o [10].

On account of elderly patients having relatively more comorbidities and displaying poorer performance, there is a tendency to use more conservative treatments like a limited EBRT irradiation volume confined to gross tumor for symptom relief. In our data, 15% of the elderly group were subject to the conservative policy, compared with only 3.6% of the younger group. In a review article, brachytherapy was not delivered for elderly patients with cervical cancer in some studies due to the consideration of toxicities, comorbidities, technical reasons, or patient refusal [9]. The other data also disclosed a similar tendency to avoid brachytherapy procedures for women aged ≥ 70 [15]. In our study, only 32.5% of patients in group 2 received ICRT application.

More contemporary RT techniques with a reduction of treatment-related side effects might be helpful for elderly patients to improve outcomes and RT completion and compliance rates. In a report, the use of RapidArc IMRT for patients aged ≥ 65 would be associated with equivalent treatment compliance, and would not result in excess toxicities [24]. The use of EBRT with IMRT, VMAT, or Tomotherapy planning with image position verification, and image-guided ICRT could all reduce RT toxicities.

Under the public health policy of preventing cancer using human papilloma virus vaccines that protect against common cancer-causing virus types, the incidence of cervical cancer in the younger age group will decline dramatically in the future. Elderly patients will occupy a greater proportion of cervical cancer patients in the future. For this reason, we made an effort to evaluate the implementation of RT and treatment outcomes in the elderly population. Tailoring the treatment strategy for elderly patients was aimed at achieving optimal outcomes and reducing complications, especially in following current precision medicine trends. Appropriate pretreatment evaluation and selection were important for elderly patients with cervical cancer receiving RT to obtain comparable or good disease control [25].

Although only 5% of newly diagnosed cervical cancer patients present stage IV disease [26], 15–61% patients will develop metastatic disease eventually, and usually within two years after first line treatment [27]. Further management for the significant proportion of patients with advanced/metastatic cervical cancer is an important issue. In our study, 32 patients in the group aged < 70 developed recurrent, persistent, or metastatic disease, and 24 of them had subsequently received systemic therapy. Nevertheless, only 5 of 13 advanced/metastatic patients in the group aged ≥ 70 were treated with systemic therapy. The most commonly used chemotherapy was platinum-based, taxane-based, or topotecan with a single or combination regimen. Only 1 patient aged ≥ 70 had received bevacizumab combining, platinum-based, chemotherapy. According to the landmark GOG 240 study, the addition of bevacizumab to either platinum or non-platinum based combination chemotherapy for advanced/metastatic cervical cancer was associated with overall survival improvement and higher response rates. In the subgroup analysis, the benefit of adding bevacizumab was more prominent for patients aged ≤ 56 [28]. The optimal systemic treatment for elderly patients still needed further exploration.

On the other hand, for patients treated with metastatic setting, there is currently no consensus for the second-line systemic therapy. In a retrospective review of recurrent/metastatic cervical cancer patients treated with first-line systemic therapy, seventy percent of patients had received more than one-line of systemic treatment [29]. However, the included patients had relatively young age. Further novel target therapy or immunotherapy with less side effects might be necessary to explore the effect of treatment for elderly patients.

There were some limitations to this study. It was a retrospective study developed at a single institution. The small number of patients in the old-aged group was correlated to heterogeneous patient characteristics, and it would be difficult to explore a more precise result in the elderly population. Further large, prospective, clinical trial is necessary to study this issue.

## 5. Conclusions

Elderly patients with cervical cancer had comparable treatment outcomes of CSS, loco-regional control, and local control rate despite receiving less comprehensive treatment, including less concurrent chemotherapy use during RT, less ICRT, less total RT dose, or limited EBRT volume. For patients aged 70 or older, especially those with a favorable stage or histopathology, a conservative treatment strategy with RT alone could be appropriate. The implementation of contemporary RT techniques for elderly cancer patients is important to reduce treatment side effects and improve compliance.

## Figures and Tables

**Figure 1 ijerph-17-04510-f001:**
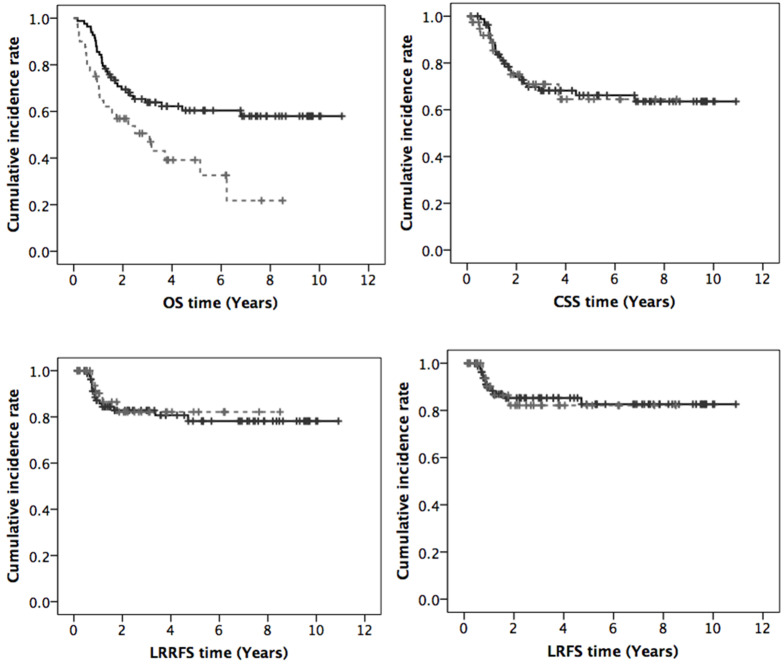
Curves of overall survival (OS), cancer-specific survival (CSS), loco-regional recurrence-free survival (LRRFS), and local recurrence-free survival (LRFS) (black line: age < 70, grey dashed line: age ≥ 70).

**Table 1 ijerph-17-04510-t001:** Patient Characteristics.

Characteristics	Age < 70 (*n* = 83)	Age ≥ 70 (*n* = 40)	*p*-Value
ECOG (%)			0.04
0–2	81 (97.6)	35 (87.5)	
3–4	2 (2.4)	5 (12.5)	
Pathology (%)			0.85
SCC	72 (86.7)	36 (90)	
Adenosquamous carcinoma	1 (1.2)	0 (0)	
Adenocarcinoma	9 (10.8)	3 (7.5)	
Small cell carcinoma	1 (1.2)	1 (2.5)	
FIGO stage (%)			0.75
IA	0 (0)	1 (2.5)	
IB	0 (0)	0 (0)	
IIA	3 (3.6)	1 (2.5)	
IIB	47 (56.6)	20 (50)	
IIIA	3 (3.6)	2 (5)	
IIIB	17 (20.5)	8 (20)	
IVA	4 (4.8)	4 (10)	
IVB	9 (10.8)	4 (10)	
Total RT dose, median (Gy) (range)	80.4 (24–106.4)	70.3 (23.4–96)	0.002
EBRT volume			0.06
Tumor only	3 (3.6)	6 (15)	
Tumor and involved regions	80 (96.4)	34 (85)	
ICRT application			<0.001
Present	58 (69.9)	13 (32.5)	
Absent	25 (30.1)	27 (67.5)	
CCRT			<0.001
CCRT	74 (89.2)	21 (52.5)	
RT alone	9 (10.8)	19 (47.5)	
OTT of RT, median (day)	68	62	0.35

Abbreviations: ECOG—Eastern Cooperative Oncology Group Performance Status, SCC—squamous cell carcinoma, FIGO—International Federation of Gynecology and Obstetrics, RT—radiotherapy, EBRT—external beam radiotherapy, ICRT—intracavitary brachytherapy, CCRT—concurrent chemoradiotherapy, OTT—Overall treatment time.

**Table 2 ijerph-17-04510-t002:** Clinical outcomes of OS, CSS, LRRFS, and LRFS.

Outcomes	Age < 70 (*n* = 83)	Age ≥ 70 (*n* = 40)	*p*-Value
OS (%)			0.006
3-year	63.9	50.6	
5-year	60.4	39.1	
CSS (%)			0.89
3-year	68.2	70.9	
5-year	66.2	64.5	
LRRFS (%)			0.75
3-year	82.8	82.1	
5-year	78.2	82.1	
LRFS (%)			0.91
3-year	85.3	82.1	
5-year	82.6	82.1	

Abbreviations: OS—overall survival, CSS—cancer-specific survival, LRRFS—loco-regional recurrence-free survival, LRFS—local recurrence-free survival.

**Table 3 ijerph-17-04510-t003:** Univariate analysis for prognostic factors of CSS in all eligible patients.

Factors	No.	3-Year CSS	*p*-Value
ECOG			<0.001
0–2	116	70.6%	
3–4	7	0%	
Age (y/o)			0.89
<70	83	68.2%	
≥70	40	70.9%	
Pathology			<0.001
SCC	108	73.7%	
Adenosquamous cellcarcinoma	1	0%	
Adenocarcinoma	12	42.8%	
Small cell carcinoma	2	0%	
FIGO stage			<0.001
I	1	100%	
II	71	82.3%	
III	30	57.1%	
IV	21	30.1%	
Total RT dose (Gy)			0.045
<80	73	62.1%	
≥80	50	77.3%	
CCRT			0.01
Yes	95	72.9%	
No	28	51.7%	
ICRT application			0.02
Yes	71	75.3%	
No	52	57.7%	
EBRT volume			0.64
Tumor only	9	80%	
Tumor and involvedregion	114	68.2%	
OTT (days)			0.58
≤56	31	65.9%	
>56	92	69.8%

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
