# Peer review of "Differences in Treatment Outcomes and Prognosis between Elderly and Younger Patients Receiving Definitive Radiotherapy for Cervical Cancer"

_ijerph, 2020, doi:10.3390/ijerph17124510_

Round 1
Reviewer 1 Report
It is an important questions to determine the optimal treatment for elderly patients with cervical cancer. In this respect the manuscript is relevant. I have though a number of comments on the Discussion:
1) Page 7, lines 183-189: This is suitable for the Introduction, not here
2) page 7, lines 189-190: I do not understand this sentence: what study do the authors refer to?
3) Page 7, lines 190-195: not relevant here, should be deleted
4) page 7, line 202: the sentence should read: The analysis of prognostic factors in our study revealed…
5) page 7, lines 206-208: “… adding chemotherapy during RT perhaps did not improve survival for old-aged patients..”. This has not been tested in the analysis. Sentence should go out, or the analysis should be made.
Author Response
Response to Reviewer 1 Comments
Point 1: Page 7, lines 183-189: This is suitable for the Introduction, not here
Response 1: I had shifted this paragraph to the Introduction part, page 2, lines 50-57.
Point 2: page 7, lines 189-190: I do not understand this sentence: what study do the authors refer to?
Response 2: The mentioned studies were related to references 13-16. However, this paragraph had been shifted to the Introduction, and the similar content was mentioned, so I deleted this sentence.
Point 3: Page 7, lines 190-195: not relevant here, should be deleted
Response 3: As another reviewer’s suggestion, I had discussed the role of chemotherapy in this paragraph. However, I had modified the content, and shift these sentences to page 7, lines 222-226, combined with other discussion of systemic therapy. I hope the modification is suitable.
Point 4: page 7, line 202: the sentence should read: The analysis of prognostic factors in our study revealed…
Response 4: The sentence was revised, page 7, line 197.
Point 5: page 7, lines 206-208: “… adding chemotherapy during RT perhaps did not improve survival for old-aged patients..”. This has not been tested in the analysis. Sentence should go out, or the analysis should be made.
Response 5: The sentence had been deleted, page 7, lines 202-203.
Reviewer 2 Report
This article compared the clinical outcomes and prognostic factors of cervical cancer between elderly and younger women. It is straightforward, well written, and concise and has clear results within the scope of a retrospective analysis. Definitely deserves to be published and is a valuable contribution to the “International Journal of Environmental Research and Public Health”. Some minor flaws could be addressed before publication.
Minor points:
[1] “Introduction”, Page 1/10, Lines 34-35:
“…and was the fourth leading cause of cancer‐related death in 2012, with an estimated 265,700 deaths.”.
The reported number of deaths should be updated. In 2018, 311,000 disease-related deaths have been reported. Please, replace reference by the following, most recent and updated.
Recommended reference: Bray F, et al. Global Cancer Statistics 2018: GLOBOCAN Estimates of Incidence and Mortality Worldwide for 36 Cancers in 185 Countries. CA Cancer J Clin. 2018 Nov;68(6):394-424.
[2] “Discussion”, Page 7/10, Lines 193-195:
“Elderly patients will occupy a greater proportion of cervical cancer in the future. For this reason, we made an effort to evaluate the implementation of RT and treatment outcomes in the elderly population.”.
This is a great comment. Based on that, it would be interesting to expand the therapeutic choices in your study beyond RT. How many patients with advanced/metastatic disease have been treated with systemic chemotherapy in each age group? Please, take into account the GOG 240 study that has investigated the addition of bevacizumab to either platinum-based or non–platinum-based combination regimen in patients with advanced or recurrent cervical cancer. The discussion section should include a paragraph with the role of chemotherapy in advanced/metastatic disease.
Recommended reference: Tewari KS, et al. Improved survival with bevacizumab in advanced cervical cancer. N Engl J Med. 2014;370(8):734-743.
[3] “Discussion”, Page 7/10, Lines 202-205:
“The analysis of prognostic factors in our study also indicated the following results. It revealed that pathology with SCC, FIGO and ICRT had significant effects on CSS, both in univariate analysis and multivariate analysis. In multivariate analysis, age, concurrent chemotherapy use, extended EBRT volume, or higher total RT dose did not shown independently influence on outcomes.”.
These results are interesting. Have you investigated the potential prognostic value of the performance status of the elderly patients in your study? In table 1, you can add the parameter of performance status (0-2 versus 3-4). This has been described as a strong prognostic factor in the literature. Please, make this comment and incorporate the following reference.
Recommended reference: Boussios S, et al. Management of patients with recurrent/advanced cervical cancer beyond first line platinum regimens: Where do we stand? A literature review. Crit Rev Oncol Hematol. 2016;108:164-174.
[4] “Discussion”, Page 7/10, Lines 226-227:
“Tailoring the treatment strategy for elderly patients was aimed at achieving optimal outcomes and reducing complications, especially in following current precision medicine trends.”.
At that point, please report that a significant number of patients present with or develop metastatic disease. Although stage IV disease accounts for only 5% of new diagnoses of cervical cancer, metastatic disease develops in 15-61%, usually within the first 2 years of completing primary treatment. So far, there is no evidence that treatment in the second-line setting improves overall survival compared with best supportive care. Please, make a statement about the retrospective review of patients with cervical cancer, treated at the Royal Marsden Hospital between 2004 and 2014. A large proportion of patients treated in the metastatic setting had received more than one line of systemic therapy.
Relevant reference: McLachlan J, et al. The Impact of Systemic Therapy Beyond First-line Treatment for Advanced Cervical Cancer. Clin Oncol (R Coll Radiol). 2017;29(3):153-160.
Author Response
Point 1: “Introduction”, Page 1/10, Lines 34-35:
“…and was the fourth leading cause of cancer‐related death in 2012, with an estimated 265,700 deaths.”.
The reported number of deaths should be updated. In 2018, 311,000 disease-related deaths have been reported. Please, replace reference by the following, most recent and updated.
Recommended reference: Bray F, et al. Global Cancer Statistics 2018: GLOBOCAN Estimates of Incidence and Mortality Worldwide for 36 Cancers in 185 Countries. CA Cancer J Clin. 2018 Nov;68(6):394-424.
Response 1: I had revised the content, and replaced reference by the updated one.
Page 1/10, Lines 34-35. Reference Page 9/10, Lines 275-276.
Point 2: “Discussion”, Page 7/10, Lines 193-195:
“Elderly patients will occupy a greater proportion of cervical cancer in the future. For this reason, we made an effort to evaluate the implementation of RT and treatment outcomes in the elderly population.”.
This is a great comment. Based on that, it would be interesting to expand the therapeutic choices in your study beyond RT. How many patients with advanced/metastatic disease have been treated with systemic chemotherapy in each age group? Please, take into account the GOG 240 study that has investigated the addition of bevacizumab to either platinum-based or non–platinum-based combination regimen in patients with advanced or recurrent cervical cancer. The discussion section should include a paragraph with the role of chemotherapy in advanced/metastatic disease.
Recommended reference: Tewari KS, et al. Improved survival with bevacizumab in advanced cervical cancer. N Engl J Med. 2014;370(8):734-743.
Response 2: The proportion of advanced/metastatic disease treated with systemic therapy in each age group was reported. I also added a paragraph to discuss the role of systemic treatment. However, as another reviewer’s suggestion, the paragraph of “Elderly patients will occupy a greater proportion of cervical cancer in the future…” should be shifted. So, I made a statement about the systemic therapy for advanced/metastatic disease, and also concurrently responded the Point 4, both on Page 7-8/10, Lines 222-250. I hope the modification is suitable.
Point 3: “Discussion”, Page 7/10, Lines 202-205:
“The analysis of prognostic factors in our study also indicated the following results. It revealed that pathology with SCC, FIGO and ICRT had significant effects on CSS, both in univariate analysis and multivariate analysis. In multivariate analysis, age, concurrent chemotherapy use, extended EBRT volume, or higher total RT dose did not shown independently influence on outcomes.”.
These results are interesting. Have you investigated the potential prognostic value of the performance status of the elderly patients in your study? In table 1, you can add the parameter of performance status (0-2 versus 3-4). This has been described as a strong prognostic factor in the literature. Please, make this comment and incorporate the following reference.
Recommended reference: Boussios S, et al. Management of patients with recurrent/advanced cervical cancer beyond first line platinum regimens: Where do we stand? A literature review. Crit Rev Oncol Hematol. 2016;108:164-174.
Response 3: I had investigated the value of performance status, and added it in table 1 (Page 4/10), table 3 with univariate analysis (Page 6/10), and multivariate analysis (Page 6/10, Lines 173-174. The discussion paragraph about performance status had been revised and incorporated the reference., Page 7/10, Lines 197-201.
Point 4: “Discussion”, Page 7/10, Lines 226-227:
“Tailoring the treatment strategy for elderly patients was aimed at achieving optimal outcomes and reducing complications, especially in following current precision medicine trends.”.
At that point, please report that a significant number of patients present with or develop metastatic disease. Although stage IV disease accounts for only 5% of new diagnoses of cervical cancer, metastatic disease develops in 15-61%, usually within the first 2 years of completing primary treatment. So far, there is no evidence that treatment in the second-line setting improves overall survival compared with best supportive care. Please, make a statement about the retrospective review of patients with cervical cancer, treated at the Royal Marsden Hospital between 2004 and 2014. A large proportion of patients treated in the metastatic setting had received more than one line of systemic therapy.
Relevant reference: McLachlan J, et al. The Impact of Systemic Therapy Beyond First-line Treatment for Advanced Cervical Cancer. Clin Oncol (R Coll Radiol). 2017;29(3):153-160.
Response 4: I had discussed the role of systemic therapy for advanced/metastatic disease, and also the second-line systemic therapy, including a statement about the reference study. All the content was on Page 7-8/10, Lines 231-250.